# Multigenerational Social Housing and Group-Rearing Enhance Female Reproductive Success in Captive Rhesus Macaques (*Macaca mulatta*)

**DOI:** 10.3390/biology11070970

**Published:** 2022-06-27

**Authors:** Astrid Rox, Sophie Waasdorp, Elisabeth H. M. Sterck, Jan A. M. Langermans, Annet L. Louwerse

**Affiliations:** 1Biomedical Primate Research Centre, 2288 GJ Rijswijk, The Netherlands; astrid.roxrox@gmail.com (A.R.); e.h.m.sterck@uu.nl (E.H.M.S.); langermans@bprc.nl (J.A.M.L.); louwerse@bprc.nl (A.L.L.); 2Animal Behaviour and Cognition, Department of Biology, Utrecht University, 3584 CH Utrecht, The Netherlands; 3Department Population Health Sciences, Unit Animals in Science & Society, Faculty of Veterinary Medicine, Utrecht University, 3584 CM Utrecht, The Netherlands

**Keywords:** reproductive success, macaques, housing, animal welfare, rearing history

## Abstract

**Simple Summary:**

It is currently common practice to house non-human primates at biomedical research facilities, e.g., rhesus macaques, in social groups. To enhance female reproductive success, peer groups are formed. In these breeding groups, infants are taken from their mother at an age of ten months and housed with animals of approximately the same age. Yet for welfare, leaving offspring with their mother and allowing multigenerational groups including families is preferred. This argues that a trade-off between female reproductive success and welfare exists. In this retrospective study we investigated the differences in female rhesus macaque reproductive success between peer groups and multigenerational groups. Our results show that females in multigenerational groups have more births per year and have higher offspring survival compared with those in peer groups. Thus, housing rhesus macaques in multigenerational groups provides a win−win situation, rather than a trade-off, in which female reproductive success and animal welfare can simultaneously be optimized.

**Abstract:**

To optimize costs and reproductive success, rhesus macaques in biomedical primate research facilities are often peer-reared. Older, dependent infants are typically removed from their natal group to enhance female reproduction. The minimal husbandry age-norm of infant removal is ten months. These practices deviate from species-specific behavior and may reduce welfare, suggesting a trade-off between female reproduction and welfare. However, the effect of breeding group type and rearing history on female reproductive success (i.e., birth rate; inter-birth interval (IBI); offspring survival) is unclear. This retrospective study investigated whether group type (i.e., peer groups versus multigenerational groups) and rearing history (i.e., peer- or hand-reared; group-reared with peer- or hand-reared mother; group-reared) affected female reproductive success in captive rhesus macaques. Data on female reproduction between 1996 and 2019 were collected at the Biomedical Primate Research Centre, Rijswijk. Birth rates were higher in multigenerational breeding groups than in peer groups. Moreover, group-reared females had higher offspring survival than peer- or hand-reared females. IBI was not affected by breeding group type or female rearing history. However, females in both peer and multigenerational breeding groups typically conceived earlier after giving birth than the husbandry infant removal age-norm of ten months. Thus, infant removal at an age of ten months does not enhance a female’s reproduction. Altogether, female reproduction and non-human primate welfare can simultaneously be optimized through multigenerational breeding groups and group-rearing.

## 1. Introduction

Group housing of captive non-human primates (NHP), compared with single housing, is widely considered an improvement of animal welfare [1,2,3,4,5,6]. Yet, group housing can be interpreted in multiple ways; e.g., it can concern peer groups (i.e., containing no matrilines) or multigenerational groups (i.e., containing at least one matriline). In peer groups, female peers live together and their offspring is removed [7,8]. A minimal legally allowed removal age of ten months is considered a golden standard, because this is suggested to optimize breeding efficiency (e.g., costs and female reproductive success) [7]. These removed individuals are often reared with peers, i.e., animals of approximately the same age are housed together. A reduction in suckling frequency is considered crucial for conception, because suckling is assumed to inhibit ovulation, suggesting that leaving infants with their mother, as in multigenerational breeding groups, may extend the inter-birth interval (IBI) relative to peer groups [9]. Moreover, infant loss shortens IBIs in Formosan rock macaques (*Macaca cyclopis*) [10] and rhesus macaques *(Macaca mulatta*) [11]. However, weaning infants at ten months of age and removing them from their maternal group deviates from the natural group dynamics. In contrast, multigenerational groups mimic the group composition and dynamics of wild macaque groups. In macaques, this concerns female philopatry, leaving offspring with their mother, resulting in multigenerational families, and male dispersal, mimicked by removal of males reaching sexual maturity from the group. Matrilines in multigenerational groups result in group stability due to the female coalitions [12]. So, multigenerational housing may enhance welfare, however, a study on peer groups reported that leaving infants in their natal group reduces female reproductive success [7]. This suggests a trade-off between welfare and female reproduction.

Removal of infants from their mother’s groups and being reared in a non-naturalistic manner can have adverse effects on NHP; peer-reared individuals express more abnormal behaviors, incompetent social behavior, and are more aggressive than group-reared individuals [13,14,15]. Peer-reared females that are removed directly after birth are often inadequate mothers themselves, suggesting that peer-rearing may lower offspring survival [16,17,18]. In line with these findings, the International Primatological Society Captive Care Committee recommends not to remove immature rhesus macaques at a young age, but to keep them with their mother for one year to 18 months, because it is unlikely that early infant removal results in improved female reproduction in seasonally breeding species [19]. 

To address the effect of group housing on female reproduction, we compared female reproductive success (i.e., birth rate, IBI, and offspring survival) of peer-housed and multigenerational-housed rhesus macaques at the Biomedical Primate Research Centre (BPRC, Rijswijk, The Netherlands). Within multigenerational groups, we also explored the effect of female rearing history on female reproductive success. Data on female reproduction were available for two periods with different husbandry regimes. Before 1996, the BPRC housed rhesus macaques in peer groups (Figure 1). After 1996, social groups were gradually (i.e., one group at a time) formed from these females, keeping offspring in the group with their mother, forming naturalistic breeding groups with at least one multigenerational matriline. We tested two opposing predictions: (1) based on the inhibition of infant suckling on female ovulation [9], a higher reproductive success in peer groups was expected. In contrast, (2) based on group stability [12], we predicted that females in multigenerational groups had a higher reproductive success than females in peer groups. Moreover, within multigenerational groups, peer- or hand-reared animals were expected to have the lowest reproductive success; group-reared animals with a peer- or hand-reared mother intermediate and group-reared animals with group-reared mother were expected to have the highest reproductive success.

## 2. Materials and Methods

### 2.1. Subjects

Data were collected on adult female (N = 358, 3–19 years of age) rhesus macaques (*Macaca mulatta*), that spent at least three consecutive years with a breeding male. This was determined by subtracting the date when the male was taken out of a group from the date when the male was introduced to a group. Female age was determined at the first breeding season she spent with an adult male. The animals were housed at the Biomedical Primate Research Centre (BPRC) in Rijswijk, The Netherlands. 

We defined peer breeding groups (N = 129) as groups with no matrilines and multigenerational breeding groups (N = 229) as groups with at least one multigenerational matriline (i.e., adult female(s) aged ≥ 3 years, with daughter(s)). In multigenerational breeding groups, females were philopatric, while related males were removed when reaching sexual maturity. The breeding male remained in the group for approximately four or five years. The few females that were on contraceptives were not included. 

Within multigenerational groups different rearing conditions were defined: R0 = peer- or hand-reared (N = 35); R1 = group-reared with peer- or hand-reared mother (N = 120); R2 = group-reared with group-reared mother (N = 74). Females that were peer-reared spent the first 12 months with their mother and were then transferred to a peer group (N = 31), except for a few hand-reared individuals that were rejected by their mother (N = 4).

### 2.2. Data Collection

Using the digital database that contained the demographic data of the colony and paper archives of the BPRC, relevant data were collected for the period 1996–2019. Between 1996 and 2003 females that were peer-housed were gradually (i.e., one group at a time) transferred to start multigenerational breeding groups, while (some) peer groups were still present. Before 1996, peer groups consisted of females and males of the same age that were reared by their mother during the first year of their life. Starting in 1996, several unrelated adult females and their offspring were introduced into new naturalistic social breeding groups together with one adult male. Until 2002 individuals were hand-reared when the mother rejected them or could not give milk. The digital database consisted of the main characteristics (e.g., date of birth) per animal housed at the BPRC, and included information on housing conditions (e.g., peer versus multigenerational groups), rearing conditions (e.g., peer- or hand-reared versus group-reared), number of births per female, age at the start of the breeding season, and age at the start of the male introduction. 

### 2.3. Ethical Statement

No ethical approval was required to perform this retrospective analysis of husbandry outcomes.

### 2.4. Measures

From the digital database, female reproductive success was determined: i.e., birth rate, IBI, offspring survival to one year, and offspring survival to three years of age. The birth rate was calculated per female by the number of births divided by the years that the female was housed with at least one adult male. Females that did not give birth were also included (peer group N = 22, multigenerational group N = 6). In peer groups, infants were removed at an age of 12 months, while in multigenerational groups females remained in their natal group and males were removed when reaching sexual maturity (i.e., around four years of age).

Average IBI in days was calculated per female by (date of 2nd birth-date of 1st birth) + (date of 3rd birth-date of 2nd birth) + (date of 4th birth-date of 3rd birth) etc./number of births. Females that gave birth ≤ 2 times were discarded for IBI calculations, resulting in N = 60 average IBIs in peer groups and N = 169 average IBIs in multigenerational groups. Time when females conceived again after giving birth was calculated by average IBI in days–166.5 days (gestation length [20]). 

Rates of offspring survival were calculated by dividing number of offspring that survived to the first year or, when they survived the first year, to three years of age by the number of births per female. Offspring of a female that were removed before reaching three years of age were removed from the dataset (N = 10 out of N = 894 offspring in total). Females that did not gave birth were omitted when calculating offspring survival, resulting in N = 107 females in peer group and N = 223 females in multigenerational group.

### 2.5. Age Interaction Maternal Age and Group Type

Note that there was a difference in female age at the start of the first breeding season with a male between peer (mean age = 8.6 ± 0.4, N = 129) and multigenerational groups (mean age = 6.3 ± 0.3, N = 229) (Wilcoxon rank sum test, W = 10277, *p*-value < 0.001). Additionally, females were significantly older at the start of the first breeding season in the peer- or hand-reared group (R0: mean age = 13.8 ± 0.5 years, N = 35) compared with the group-reared females (R1: mean age = 5.4 ± 0.2 years, N = 120; R2: mean age = 4.1 ± 0.1 years, N = 74) (Kruskal−Wallis test X^2^ = 98.6, *p*-value < 0.001, peer N = 129, multigenerational N = 229; Benjamini and Hochberg post hoc comparison of R0-R1 *p*-value < 0.001, R0-R2 *p*-value < 0.001, R1-R2 *p*-value < 0.001). However, we found no effect of age on female reproductive success (Appendix A).

### 2.6. Statistics

Measures of female reproductive success were not normally distributed (Shapiro−Wilk normality test: female reproduction success: *p* < 0.001; IBI: *p* < 0.001; offspring survival to first year: *p* < 0.001; offspring survival to three years of age: *p* < 0.001). First, to compare peer versus multigenerational breeding groups (N = 129 respectively N = 229), a Wilcoxon rank sum test was performed. Second, to compare rearing history conditions (R0: peer- or hand-reared, R1: group-reared with peer- or hand-reared mother, R2: group-reared with group-reared mother), a Kruskal−Wallis test was performed with a Benjamini and Hochberg post hoc test. A *p*-value ≤ 0.05 was considered significant.

## 3. Results

First, the effect of breeding group type on reproductive success of females was explored. Females in multigenerational breeding groups produced a significantly higher number of offspring per female per year (0.74 ± 0.02 offspring per female per year, N = 229) than females in peer groups (0.47 ± 0.03 offspring per female per year, N = 129) (Wilcoxon rank sum test, W = 7910, *p* < 0.001; Figure 2a; Appendix A). However, IBI did not differ between females in peer (IBI = 448 ± 16 days, N = 60) and multigenerational (IBI = 425 ± 7 days, N = 169) breeding groups (Wilcoxon rank sum test, n1 = 169, n2 = 60, W = 5529, *p*-value = 0.30) (Figure 2b). 

Second, the survival of offspring was investigated in relation to breeding group type. Overall offspring survival was 80% (243 offspring survived from a total of 304 births) to one year of age and 76% (231 offspring survived from a total of 304 births) to three years of age in peer groups, respectively 90% and 85% (806 respectively 763 offspring survived from a total of 894 births) in multigenerational groups. Offspring survival per female to both one and three years of age was significantly higher in multigenerational breeding groups (average offspring survival per female: 0.90 ± 0.20 respectively 0.86 ± 0.01, N = 223) compared with peer breeding groups (average offspring survival per female: 0.77 ± 0.34 respectively 0.74 ± 0.03, N = 107) (Wilcoxon rank sum test, offspring survival to one year of age: W = 13909, *p*-value = <0.01; offspring survival to three years of age: W = 13768, *p*-value = 0.01) (Figure 2c,d). 

Finally, within multigenerational breeding groups, the effect of rearing history on female reproduction was determined. Female rearing history did not affect birth rate and IBI (Kruskal−Wallis test, birth rate X^2^ = 3.0, *p* = 0.23; IBI X^2^ = 1.36, *p*-value = 0.51) (Figure 3a,b). However, offspring survival to one and three years of age was higher in group-reared females (i.e., R2) compared with peer- or hand-reared females (i.e., R0) (offspring survival to one year of age: Kruskal−Wallis test, X^2^ = 6.85, *p* = 0.03; Benjamini and Hochberg post hoc comparison of R0-R1 *p*-value = 0.06, R0-R2 *p*-value = 0.04; offspring survival to three years of age: Kruskal−Wallis test, X^2^ = 7.42, *p*-value = 0.02; Benjamini and Hochberg post hoc comparison of R0-R1 *p*-value = 0.07, R0-R2 *p*-value = 0.03) (Figure 3c,d). There was no difference in offspring survival between group-reared females that had peer- or hand-reared mothers (i.e., R1) or group-reared mothers (i.e., R2) (Kruskal−Wallis test: offspring survival to first year of age between R1-R2: Kruskal−Wallis with Benjamini and Hochberg post hoc comparison: *p* = 0.37; offspring survival to three years of age between R1-R2: Kruskal−Wallis with Benjamini and Hochberg post hoc comparison: *p* = 0.19). 

## 4. Discussion

This retrospective study investigated the effect of breeding group type and rearing history on female reproductive success in captive rhesus macaques. We had two opposing predictions: (1) based on the inhibition of female ovulation due to infant suckling [9], a higher reproductive success in peer groups was predicted. In contrast, (2) based on group stability [12], we predicted a higher reproductive success in multigenerational breeding groups. Consistent with the second prediction, birth rates and offspring survival were higher in multigenerational groups compared to peer groups, although IBI was not affected by breeding group type. Moreover, offspring survival was higher for group-reared females than in peer- or hand-reared females, yet within multigenerational groups both birth rate and IBI were not affected by female rearing history. Thus, multigenerational groups that enhance welfare [21], also enhance female birth rate and offspring survival.

### 4.1. Breeding Group Type

First, we analyzed the effect of breeding group type on female reproductive success. Supporting the hypothesis that multigenerational groups would benefit female reproduction, we found that females housed in multigenerational groups had a higher number of births per year compared with females housed in peer groups. Yet IBI was not affected by breeding group type. The different outcomes of average birth rate and IBI were not expected. Based on increased birth rates in multigenerational breeding groups, it was expected that IBIs would decrease; more births per year may result in shorter IBIs. However, these measures of female reproductive output may be different due to differences in the underlying distribution of the data: while the average birth rate over several years can take many intermediary forms, due to the seasonal breeding in rhesus macaques the IBI is either around 12 or 24 (or more with increments of 12) months. Thus, more accurate estimates for reproductive rate may be derived from the average birth rate than from IBI. Similarly, rearing history and early infant removal did not increase IBI in rhesus macaques [8]. Thus, birth rate can be improved by housing female rhesus macaques in multigenerational groups instead of peer groups, while IBI is not affected by breeding group type, rearing history (our study) nor infant removal age [8].

A second important component of female reproductive success concerns offspring survival, which may depend on group type. As expected, offspring survival to one and three years of age was higher in multigenerational breeding groups compared with peer breeding groups. Increased offspring survival is in agreement with a higher offspring viability in group-housed compared with solitary-housed female rhesus macaques [22]. Pair-housed rhesus macaques also have a higher rate of infant survival and higher infant weight gain than single-housed females [5]. These effects of the group types may differ, because group types differ in group composition. In natural (multigenerational) groups the presence of sires and other pregnant females increase female reproductive success, as was found in pigtailed macaques (e.g., viable birth, gestation length) [23]. Moreover, increased group sizes lead to a decreased number of females needing clinical treatment due to injury and a decreased gestation length [23]. However, the behaviors mediating these effects is unclear. In addition, multigenerational groups are more stable than peer groups with no matrilines [12,24], which may decrease aggression and therefore stress levels. Indeed, increased social stressors can lower reproductive success [25]. Thus, multigenerational groups may be advantageous over peer groups due to differences in group composition and its (potential) effect on multiple social behaviors. Altogether, housing females in multigenerational groups enhances birth rate and offspring survival relative to peer groups.

### 4.2. Maternal Rearing History

In addition to breeding group type, we investigated the effect of female rearing history on female reproductive success. Against our expectation, within multigenerational groups female rearing history did not affect birth rate. This indicates that early experiences during infancy of peer-reared females (i.e., rearing history) are not the major factor determining female birth rate, whereas the actual housing situation (i.e., breeding group type) does affect birth rates. So, switching to multigenerational housing improves the birth rates of all females, including peer-reared females.

In contrast, offspring survival does depend on female rearing history: peer-reared females (R0) had lower offspring survival than group-reared females (R2). Offspring survival may be an indicator of individual maternal competence [26]. Social competence of socially reared individuals is higher compared with that of peer-reared individuals removed at an age of 12 months [13,14] and 2–3 months [26]. Similarly, peer-reared female rhesus macaques that were separated from their mother at an age of 12–24 months have a lower offspring survival rate to 30 days compared with group-reared females [26]. This may result from maternal (in)competence or from increased infant kidnapping or abandonment by peer-reared females [25]. Offspring of peer-reared mothers are more frequently held in an abnormal ventral position (upside down or facing to the outside of the abdomen) compared with offspring of group-reared mothers [26]. This may result in lower offspring survival, due to blocking of access to the mother’s breast. In addition, infants reared in peer groups are touched more often by group members than infants reared in group, who are groomed more by their own mothers [26]. Altogether, maternal rearing history possibly influences maternal competence and, accordingly, within multigenerational groups, group-reared females have higher offspring survival than peer- or hand-reared females.

Against our expectations, among group-reared females there was no difference in offspring survival rate between R1 (i.e., group-reared females with peer- or hand-reared mothers) and R2 (i.e., group-reared females with group-reared mothers). A possible explanation for this finding is female rhesus macaques that were exposed to suboptimal maternal care (e.g., abuse and/or maternal rejection) during infancy express a higher interest in infants than females that obtained adequate maternal care [27]. Since the offspring of peer- or hand-reared females resided in a group with infants, they may have compensated for their mother’s lower maternal competence. So, within multigenerational groups, peer- or hand-rearing of a mother does not seem to influence the offspring survival of their daughter(s).

Altogether, our data imply that housing rhesus macaques in multigenerational groups is not only preferred from a welfare point of view [28], but is also advised when high female reproductive success is encouraged.

### 4.3. No Trade-off between Female Reproductive Success and Welfare

For breeding facilities, there seems to be a trade-off between female reproductive success and welfare. Peer groups are expected to enhance female reproductive success [7], while multigenerational groups are expected to enhance welfare [12]. Opposing the first expectation, we found that multigenerational groups enhance female reproductive success over that of peer groups. This contrasts with the assertion that suckling from infants suppresses ovulation and will lengthen the IBI [7,9]. Moreover, the average birth rate of 0.74 ± 0.02 in multigenerational group-housed females (i.e., no offspring removal) was similar to 0.74 at an infant removal age of six months in peer groups [7]. This suggests that multigenerational groups result in a similar female reproduction as groups where infants are removed at an age that is younger than what is currently accepted (i.e., ten months of age; [8]). To explore this further, we transformed the data on IBI (from Figure 2b), which did not differ between the two breeding group types, to the age of the previous infant when a female conceived again (Figure 4). This shows that for both peer-housed and multigenerational group-housed rhesus macaques at the BPRC, the average time of conception was shorter (peer-housed females: mean of 9.3 ± 0.5 months respectively multigenerational females: mean of 8.6 ± 0.2 months) than the legally allowed age of infant removal (i.e., ten months) [8]. Thus, our data indicate that both peer-housed and multigenerational group-housed female rhesus macaques are already pregnant when their previous infant reaches the age of ten months. Although our data, that were systematically and empirically collected in different periods, demonstrate that infant removal at ten months of age will not enhance inter-birth intervals in either group type. 

Our outcomes counter intuitions on the effect of suckling on female reproduction. However, at four months of age, infant suckling frequencies decrease steadily and they obtain a considerable portion of their diet from solid food [9,29]; also, in the wild it has been observed that infants remain close to their mother, even when the subsequent baby is born [30,31]. This suggests that suckling behavior itself does not necessarily reflect milk intake, but may also have a different function. For example, the infant may obtain comfort and social support from the mother, even when it does not gain much milk. This may explain why infant removal at an age of ten months does not speed up its mother’s ovulation, and thus fertility. Moreover, removing an infant even younger (at six months of age [7]) does not enhance the birth rates relative to the BPRC multigenerational group housed female rhesus macaques (see above). Our findings strongly indicate that infant removal does not enhance female reproductive success.

## 5. Conclusions

In conclusion, removal of infants at an age of ten months does not enhance female reproductive success. More importantly, females in multigenerational groups have higher reproductive success and offspring survival than those in peer groups, and multigenerational groups are therefore preferred. Our results suggest that, against old sentiment, no trade-off exists between female reproductive success and animal welfare. So, housing rhesus macaques in multigenerational groups provides a win−win situation for both female reproductive success and animal welfare. 

## Figures and Tables

**Figure 1 biology-11-00970-f001:**
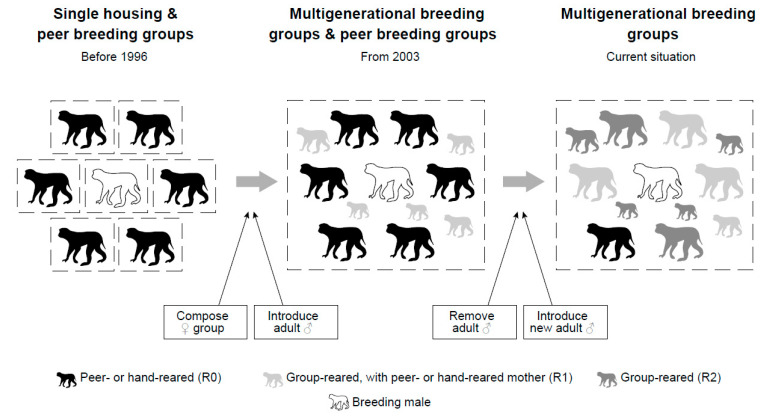
Transition of the different types of housing conditions in which rhesus macaques were previously (<1996) and are currently kept in the BPRC (Rijswijk, The Netherlands). After 1996, the breeding groups consisted of multigenerational groups with varying rearing history (black: peer- or hand-reared (R0); light gray (R1): group-reared, with peer- or hand-reared mother; dark gray: group-reared (R2); white: breeding male).

**Figure 2 biology-11-00970-f002:**
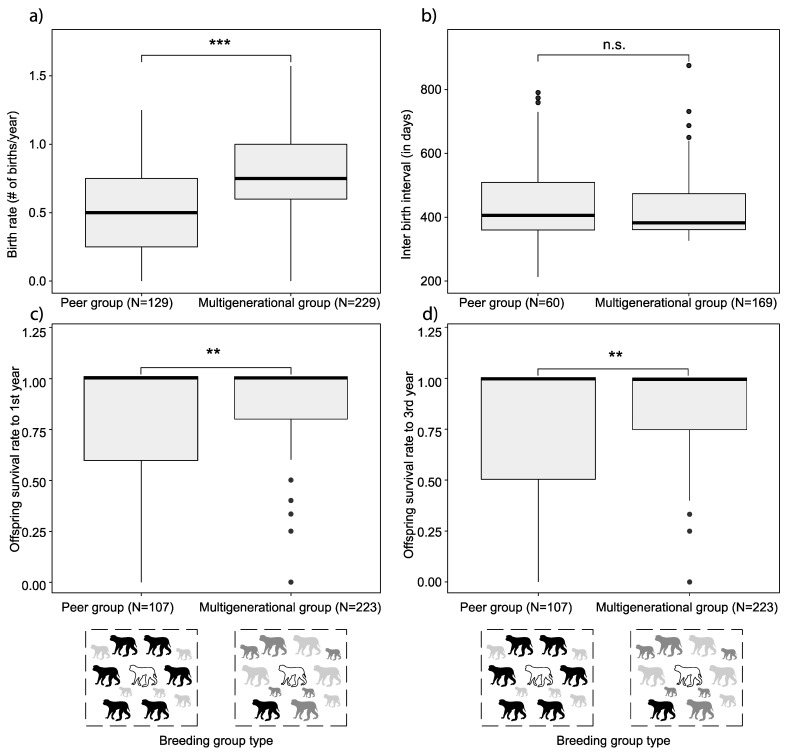
Birth rates, IBIs and rates of offspring survival per breeding group type: (**a**) birth rates were significantly higher in multigenerational breeding groups compared to peer breeding groups in captive rhesus macaques; (**b**) IBIs did not differ between peer and multigenerational breeding groups; (**c**) offspring survival to one year and (**d**) three years of age was significantly higher in multigenerational breeding groups compared to peer groups. ** indicates *p* ≤ 0.01; *** indicates *p* ≤ 0.001.

**Figure 3 biology-11-00970-f003:**
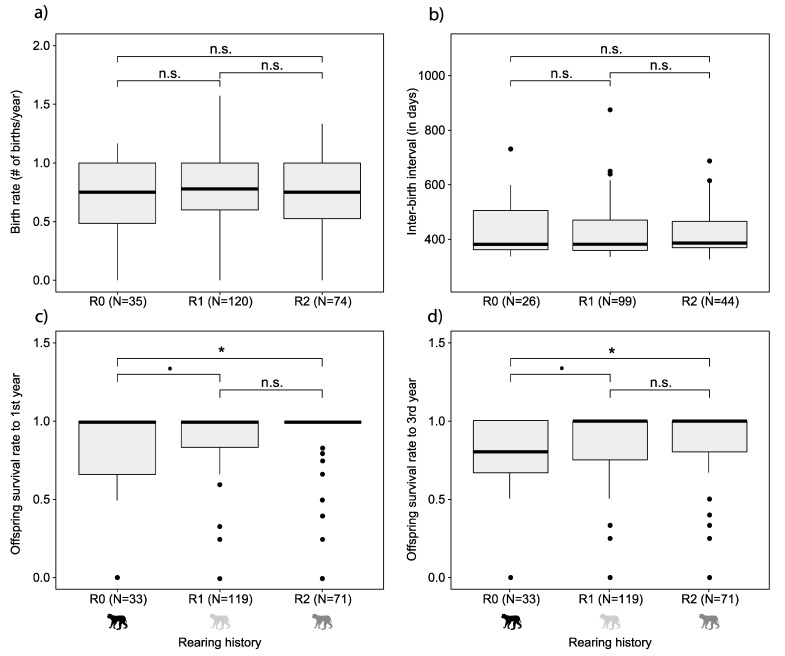
Birth rates, IBIs and offspring survival per rearing history, within multigenerational breeding groups: (**a**) birth rates were not affected by female rearing history in captive rhesus macaques (R0 = peer- or hand-reared; R1 = group-reared with peer- or hand-reared mother; R2 = group-reared); (**b**) IBIs were also not affected by female rearing history; (**c**) offspring survival to first year and (**d**) three years of age was significantly lower in R0 compared with R2. * indicates *p* ≤ 0.05; • indicates 0.05 < *p* < 0.10.

**Figure 4 biology-11-00970-f004:**
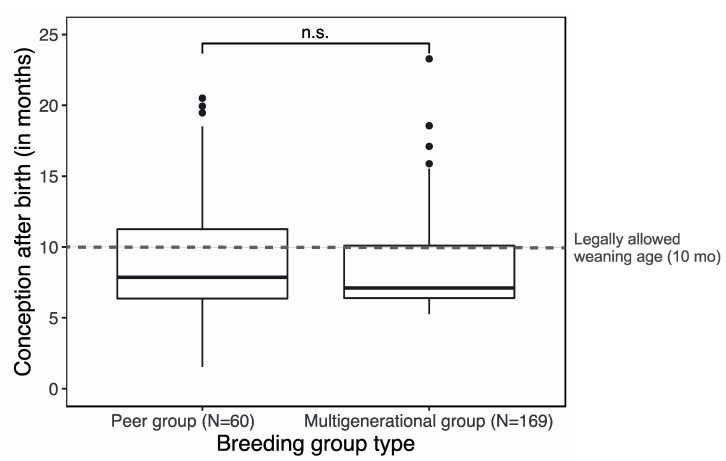
Females in peer and multigenerational breeding groups conceived earlier after giving birth (peer-housed females: 9.3 ± 0.5 months respectively multigenerational females: 8.6 ± 0.2 months) than the legally allowed removal age norm of ten months for transferring dependent offspring to peer groups in captive rhesus macaques. Note that the data from this figure are derived from Figure 2b.

## Data Availability

Data are available on request.

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
