# Peer review of "Multigenerational Social Housing and Group-Rearing Enhance Female Reproductive Success in Captive Rhesus Macaques (Macaca mulatta)"

_biology, 2022, doi:10.3390/biology11070970_

Round 1

Reviewer 1 Report

Review of “Multi-generational social housing and group rearing enhance female reproductive success in captive rhesus macaques”.

The authors aim to determine whether there is a trade-off between animal welfare and reproductive output when lab-housed macaques are housed naturalistically, in multi-generational groups, or when housed in peer-groups. The authors used 25 years’ worth of data collected as a research laboratory to determine the effects of maternal rearing on the inter-birth interval, offspring survival, and the birth rate. The authors found that reproductive output did not decrease in multi-generational groups, and therefore recommend that lab-housed macaques are managed in more naturalistic social groups. This paper was an enjoyable read, and I would suggest that only minor revisions need to be carried out. There are a few sections that were a bit more difficult to read which I outline below.

Line by line comments:

L34: was > were

L51: are there males in peer groups?

L51-52: do group sizes and age:sex ratios reflect those in the wild in multi-generational groups?

L65: enlarge > extend

L65-66: it is not clear to me how the sentence on nipple contacts relates to the rest of the paragraph. Please expand on the relationship between nipple contacts and female reproductive success.

L77: Your argument about improved breeding efficiency doesn’t have as much impact after the discussion about lower offspring survival. Are there any references that state that the cost of decreased survival is outweighed by theoretically shorter IBIs?

L78: Does any form of reproductive management occur in the groups e.g. contraception or culling, or are animals just free to breed?

L81: How do rates of reproductive success in the centre compare to that of the wild? Often in captivity, IBIs, age of first conception are shorter than in the wild. I would appreciate an expanded explanation of what naturalistic is in the context of a rhesus macaque. What do wild groups look like? What is the age of dispersal? What are in situ reproductive patterns like?

Figure 1:

There is an issue with the symbols in the figure (in the boxes with ‘Compose ___’, ‘Introduce adult ___’ etc.).

What does the white macaque represent? Include it with the other explanations.

L101: Are breeding males typically peer or multi-generationally reared? Was the breeding male included as a factor contributing to reproductive success? Were all males proven breeders? What are causes of offspring mortality? Could male or female aggression have contributed, particularly if they're abnormally reared?

L105: why did you chose 3 years?

L114-115: You have already mentioned this in L105.

L120: were they moved at 10 or 12 months?

L129: what happened to rejected offspring after 2002?

L139: was it assumed that all individuals were fertile?

L140: don’t need the IBI abbreviation again.

L153: does this include all individual removed for peer groups at 10 months?

L160 + further forward (ff): for clarity when reading, it’s easier to read P-values as <0.1, <0.01, <0.001 etc. rather than Xe-X.

L173: “and 0.05> p <1.00 was considered a non-significant trend” is not needed.

L176: reproduced > produced

L197+ff: just 'offspring survival' (remove rates) or say 'rates of offspring survival'

L202: this sentence doesn’t read well. I would rephrase to "If offspring survived the first year, they were likely to survive to age three' or similar.

L202-204: these sentences should follow information on offspring survival (L182-186).

Figure 2: the images of group composition aren’t needed as you explain the group type in the x-axis.

Figure 2 legend: a repeat of the results is unnecessary in the figure legend.

Figure 3: again, images of macaques aren’t needed as you explain the rearing in the x-axis.

L232: with > to

L234: compared with > than in

L236: I don’t believe that reference 8 directly shows that multi-generational groups improve welfare, rather it looks at male emigration. Reference 1 in the Rox et al. reference does.

L241-242: “and countering the proposition that peer-groups would enhance female reproduction” is unnecessary.

L250: steps > increments

L258: Include a reference for this statement.

L260-272: This paragraph is quite disjointed as it reads like you keep remembering things to add on.

L266: why do they need clinical treatment? Is this due to injury?

L269: stress levels > social stressors.

L272: enhances birth rate and offspring survival relative to peer groups.

L275: next to > in addition to

L280: benefits > improves

L287: delete offspring. Replace to with of.

L291: is this due to maternal inexperience? If so, make sure the argument follows L287-289.

L307: group-housing

L316-318, L320-330, and Figure 4: avoid including actual results in the discussion. These should be in the results.

L318-320: This is really difficult to understand. Is your argument that offspring removal had no effect on birth rates?

L321: how did you transform the data?

L332: delete ‘s’ from infants

L349: could > does

Reviewer 2 Report

The authors present a highly important analysis of female reproductive success in the setting of peer groups versus multi-generational groups. The relevance of this work is high, considering that weaning practices/breeding logistics may remove animals as soon as 10m, which introduces welfare costs, as early weaning is not species typical behavior. 

Large cohort data, from a single site – which excludes the influence of other variables, are critically important in challenging dogma that exists or anecdotal practices in NHP breeding. While the welfare implications have been explored and peer groups present challenges, it is accepted in some situations as it relates to breeding efficiency. The authors do an expert job in providing evidence based support for equivalent (or superior) markers of breeding efficiency in multi-generational groups. 

The authors also bring in the interesting component of hand-rearing into this retrospective analysis.  It is quite novel to show that reintroduction into a multi-generational groups can effectively blunt factors associated with peer-rearing.

The authors should add a sentence or two in the discussion relating to limitations of the study, for example, the retrospective nature introduces the concern that evaluations were performed in essentially different ‘eras’.

Introduction is well rounded and establishes the significance of this issue. Figures are clear and excellent.  Discussion is well constructed and balanced. Congratulate the authors on an excellent contribution to the field.

Reviewer 3 Report

This is a very interesting paper that contains extremely important data for those interested in the production of rhesus monkeys in captive breeding colonies.  The comparison between breeding groups formed with females from a small range of ages (‘peer’ groups) and breeding groups composed of females and their subsequent offspring (‘multi-generational’ groups) is an extremely relevant one.  The findings of this study, that multi-generational groups ‘produce’ better (more births and higher rates of survival) than peer groups, should be of importance to every captive colony that is attempting to optimize production and welfare of macaques (particularly for rhesus, but the findings are probably applicable to long-tailed and pig-tailed macaques as well).  The findings from this study empirically and systematically address an issue that has needed, but has received little, attention in the past, and thus need to be published.

While the results of this study are extremely important and the results themselves are relatively well presented, there are other parts of the manuscript that are somewhat confusing and that need to be clarified in order to maximize the positive impact of this paper. 

A clear definition of peer-reared needs to be provided.  In some of the literature cited, especially the older literature from Harlow and his colleagues, peer-reared refers to something quite different from what is meant by peer-rearing at the BPRC.

Weaning probably needs to be more explicitly defined as well – by 10 months of age, females have probably already weaned (from breast feeding) their infants; what occurs at 10 months by humans is removal from the natal group, rather than ‘physiological’ weaning.

The data and presentation on rearing history are considerably more confusing than the data on breeding group type.  The authors need to clarify the explanation of rearing history types or even consider eliminating it as a ‘variable’ in this study.

As a seasonally reproducing species, with IBIs in the neighborhood of 365-400 days, it is not surprising that females conceived again, prior to their ‘current’ offspring reaching 10 months of age.  The thought process that resulted in weaning at 10 months was intended to shorten IBIs, which the data from this paper clearly demonstrate does not occur.

Specific suggestions to improve the overall presentation are provided below.

Introduction

Lines 46-67         These two paragraphs do not flow particularly well, both within and between paragraphs.  The authors should have another look at these paragraphs.

Lines 68-78         The references cited in this paragraph refer to a type of peer rearing that was practiced long ago and is quite different from the peer groups described in the present study.  Lines 76-78 don’t really fit in this paragraph.

Line 104               Strong sample size

Methods

Lines 124-126     The ‘description’ of solitary housed females probably needs additional explanation.

Line 152               Again, strong sample size

Lines 154-163     If female rhesus gave birth for the first time at 4 years of age, then it seems unlikely that many R1 and R2 females (mean ages were 5.4 and 4.1 years, respectively) could have been included in the study (needing to have given birth at least 3 times).  Additional explanation is required here.

Results

Lines 174-181     The Ns used for the IBI calculations seem too small.  If 894 offspring were included in the analysis, shouldn’t there be more than 60 + 169 IBIs from a total of 358 females?

Lines 185-186     Infant survival rates to one year of age in peer breeding groups seem problematically low.  Do the authors have any ideas concerning why this might be?  Clearly, multi-generational groups were necessary to keep this colony going.

Discussion

Lines 226-237     Excellent summation of the very important results of this study.

Lines 286-290     This is another point in the manuscript where the authors should make sure that the comparisons that they are making are relevant to their definition of peer breeding groups.

Lines 310-330     This may be the most important paragraph in the discussion, because it dispels the ‘myth’ that removal of infant rhesus from their mother enhances productivity.  It is important for the authors to emphasize that the data that they systematically and EMPIRICALLY collected disproves that ‘early’ removal of infants from dams is necessary to support adequate production.

Line 341               “plead” is not the correct word here.  The empirical data have clearly demonstrated that early infant removal does not achieve the goals that it was thought to achieve. This statement must be stronger.  “The data strongly indicate…”

Lines 348-355     The conclusions are strong and clearly supported by the data.

References

Some of the older references, especially those that include Harlow, are not really relevant to this paper, and it might be better to delete them.  It might be useful to cite some of the recent social network analysis papers published by McCowan and her group to address social dynamics in multi-generational rhesus breeding groups.

Overall, the data in this paper are REALLY important and need to be published.  If the authors can address the issues related to the clarity of the presentation and appropriate definitions of some of their critical terms, then this should be a highly cited paper. 
